# The Research Progress on Immortalization of Human B Cells

**DOI:** 10.3390/microorganisms11122936

**Published:** 2023-12-07

**Authors:** Huiting Xu, Xinxin Xiang, Weizhe Ding, Wei Dong, Yihong Hu

**Affiliations:** 1Pediatric Department, Nanxiang Branch of Ruijin Hospital, Jiading District, Shanghai 201802, China; tingtingdoctor@sina.com; 2CAS Key Laboratory of Molecular Virology & Immunology, Institutional Center for Shared Technologies and Facilities, Pathogen Discovery and Big Data Platform, Shanghai Institute of Immunity and Infection, Chinese Academy of Sciences, Yueyang Road 320, Shanghai 200031, China; 15580231023@163.com (X.X.); weizheding@outlook.com (W.D.); 3Hengyang Medical College, University of South China, Hengyang 421200, China; 4Peking-Tsinghua-NIBS Joint Program, School of Life Sciences, Tsinghua University, Beijing 100084, China; 5University of Chinese Academy of Sciences, Beijing 100049, China

**Keywords:** human B cell, EBV, SV40, genetic modification, monoclonal antibodies

## Abstract

Human B cell immortalization that maintains the constant growth characteristics and antibody expression of B cells *in vitro* is very critical for the development of antibody drugs and products for the diagnosis and bio-therapeutics of human diseases. Human B cell immortalization methods include Epstein-Barr virus (EBV) transformation, Simian virus 40 (SV40) virus infection, *in vitro* genetic modification, and activating CD40, etc. Immortalized human B cells produce monoclonal antibodies (mAbs) very efficiently, and the antibodies produced in this way can overcome the immune rejection caused by heterologous antibodies. It is an effective way to prepare mAbs and an important method for developing therapeutic monoclonal antibodies. Currently, the US FDA has approved more than 100 mAbs against a wide range of illnesses such as cancer, autoimmune diseases, infectious diseases, and neurological disorders. This paper reviews the research progress of human B cell immortalization, its methods, and future directions as it is a powerful tool for the development of monoclonal antibody preparation technology.

## 1. Introduction

Cell immortalization refers to cells that grow *in vitro* induced by or under the influence of external factors and whose growth cycle is different from that of normal cells, and the trend of senescence is avoided in order to have high proliferation and long passage [1]. Immortalization is a necessary stage in the process of malignant transformation from normal cells to cancer cells, so cell immortalization has become a research hotspot related to cancer. Meanwhile, the immortalization of B cells is an essential part of immunotherapy for cancer and one of the sources and bases for preparing specific monoclonal antibodies. They support the production of rare antibodies and can overcome the disadvantages of other production methods, such as the human naive library combined with a phage display platform [2,3,4] or the single B cell platform [5,6], through screened monoclonal antibodies. Monoclonal antibodies of the same species do not cause immune antagonism and play an indispensable role in the treatment of diseases. Generally, the frequency of cell immortalization is very low, which is less than 1 × 10^−12^ in human cells and 1 × 10^−6^–1 × 10^−5^ in rodent cells. Therefore, the spontaneous immortalization of animal cells is scarce [7].

Currently, artificial methods are mainly used to induce cell immortalization. The immortalized mechanisms reported so far include radioactive mutations, telomere and telomerase activation, viral transformation, activation or inhibition of oncogenes and tumor suppressor genes, etc. [8]. Although the mechanisms of cell immortalization are similar, the methods of immortalization are varied for different cells. The activation of human telomerase reverse transcriptase (hTERT) is an important step in cell immortalization [9,10,11], as the stable expression of the hTERT gene in primary cultured cells can stabilize the length of telomeres and immortalize the cells of different species [12,13]. However, hTERT overexpression has never immortalized human B cells successfully. While cell lines transformed by Abelson murine leukemia virus (v-Abl cells) were first reported and used for studying the regulation of B cell development [14], v-Abl blocks B cell development, resulting in B cells failing to mature and produce antibodies [15]. Shortly afterward, Köler and Milstein discovered the immortalization of the murine B cells method—the hybridoma technology generating large amounts of monoclonal antibodies—which was a revolution in immunology [16]. The immortalization of mouse B cells by fusing them with myeloma cells has since become widely practiced. Hybridomas based on B cells of other rodents (rat hybridomas are common) are very abundant and widely commercially available [17]. In contrast, the immortalization of human B cells (and of non-rodent species) proved to be more challenging. There are no human myeloma cell lines that proved to be successful fusion partners. “Hetero-hybridomas” created by the fusion of human B cells with mouse myeloma cells were genetically unstable and could not be stably maintained in culture. So, it is necessary to look for efficient immortalization approaches for human B cells.

Here, we reviewed approaches for human B cell immortalization discovered in those decades. The typical methods include Epstein-Barr virus (EBV) transformation, Simian virus 40 (SV40) infection, *in vitro* genetic modification, and activating CD40 signal [8]. It is convincing that spontaneously immortalized B cells activated by exogenous signals improve the simplicity and safety of their application. In this review, these leading immortalization strategies are discussed in detail, evaluating their advantages, restrictions, and further potential for the future contribution to the diagnosis and therapeutics of diseases. Hopefully, with high immortalization efficiency, easy-to-screen, and greater stability characteristics, combined with advanced technologies such as microfluidic platform, single cell analysis, fluorescent spectrometry, and cell surface fluorescence immunosorbent assay, etc., fully developed B cell immortalization technology in the future will provide an excellent method for the isolation of antigen-specific B cells that produce humanized antibodies for medical purposes with high efficiency.

## 2. Immortalize Human Peripheral Blood B Lymphocytes by Epstein-Barr Virus

Epstein-Barr virus (EBV), a human herpesvirus causing infectious mononucleosis, is highly immunogenic having a more than 95% seropositive ratio in the world community and is associated with Burkitt lymphoma and nasopharyngeal carcinoma in etiology [18,19]. It contributes to 200,000 cancers per year around the world [20], and after the establishment of latent infection in B lymphocytes, EBV can persist in the human host for life [21]. EB virus transformation is a standard method to immortalize isolated human B cells [22]. EBV can transform isolated mononuclear leukocytes, which contain human peripheral blood B lymphocytes, into immortal human lymphoblastoid cell lines (LCLs) in an appropriate *in vitro* environment, and their biochemical and molecular biological characteristics remain unchanged [23,24]. The EBV genome is a linear double strand of about 172 kb containing more than 85 genes, but only a few are expressed in EBV-infected B lymphocytes, called latent genes [24,25]. Latent proteins can activate the interaction between cell growth factor and its receptor, and change the life cycle of B lymphocytes, thus immortalizing the cells.

Six latent virus genes, including Epstein-Barr nuclear antigen LP (EBNA-LP), nuclear antigen 3A (*EBNA3A*), nuclear antigen 3C (*EBNA3C*), nuclear antigen 1 (*EBNA1*), nuclear antigen 2 (*EBNA2*), and latent membrane protein 1 (*LMP1*) [26], are mainly expressed during the transformation of B cells into LCLs by EB virus [27]. *EBNA1* and *EBNA2* are the keys to inducing the immortalization of B cells, while *EBNA3A*, *EBNA3C*, and *LMP1* are also involved in the *in vitro* transformation of B cells. *EBNA1* is a homologous dimer DNA-binding protein that can bind to multiple sites in the host genome [28]. *EBNA1*′s expression in the latent and release periods of viruses mediates replication and segregation of the respective viral genomes [29,30], which is crucial for maintaining replication termination at OriP and viral episome maintenance [31]. *EBNA2* is a primary viral transcription activator and can up-regulate the expression of host genes and other EBNA genes [32]. It activates the expression of viral oncoprotein *LMP1* during the initiation and maintenance of B cell immortalization [33], which accelerates the immortalization of infected B cells into LCLs, enabling them to proliferate indefinitely during culture. Thus, the *LMP1* oncoprotein is essential for the continued growth and survival of LCLs [34]. It simulates a pro-survival tumor necrosis factor receptor, which constitutively sends out signals through the NF-KB pathway to promote the proliferation of transformed cells and inhibit apoptosis [35]. Inhibition of the activated NF-KB pathway downstream of *LMP1* resulted in the apoptosis of LCLs. However, studies have shown that B cells still proliferate rapidly with *LMP1* expression and low NF-KB activation levels in the early post-infection period [35,36]. In early infected B cells, the expression of *EBNA3A* protein can induce the resistance to apoptosis of B lymphocytes and primary B cells by down-regulating the Bim gene [37], thus reducing the “starting power” of infected cells to apoptosis. So, these early infected cells showed no apparent apoptosis, although the cell DNA damage response was strongly activated [38]. In addition, the EBN3A family epigenetically down-regulates other human genes involved in cell cycle regulation [39]. For example, the expression of two host proteins, *MCL-1* and *BFL-1*, is controlled by *EBNA3A*. Meanwhile, *EBNA3C* and *EBNA3A* inhibit the expression of tumor suppressor p16 in combination to maintain the continuous proliferation of B cells and transform B cells into immortalized LCLs [40,41]. In summary, the EBV latent gene mimics the critical factors of natural B cell growth, promotes its proliferation to form immortalized LCLs, and inhibits host innate tumor suppressor responses to uncontrollable proliferation (Figure 1).

Studies have found that another mechanism by which B lymphocytes are transformed by EBV may be related to the inhibition of the p53 tumor suppressor and the enhancement of telomerase activity [42]. P53 induces apoptosis by blocking normal cells’ transition from G1/G0 to S phase. Although the EBV’s transformation of LCLs elevated p53 in cells, the EBV latent proteins and cell proteins induced by the EBV block p53 protein-mediated apoptosis so that the LCLs become immortalized [43]. On the other hand, studies have found that telomere elongation is activated, both in the early and late stages, after EBV infection [44]. Telomere length is maintained with significantly increased telomerase activity, and B cells break away from the M1 and M2 phases of LCLs transformed by EBV [41]. Moreover, the EBV transformation of LCLs is a reversible process without a complex artificial tumor background, so it is a good model for further study of cell immortalization.

A series of factors affecting the transformation of B cells into immortalized LCLs by EBV have been found [45]. Using TLR9 agonists improves the immortalization efficiency of human memory B cells [46]. Gisela Nogales-Gadea et al. studied the optimal conditions for B cell transformation and established an effective LCL culture method [47]. The best culture condition is the RPMI 1640 culture medium containing 20% FBS at 37 °C with 5% CO_2_ [47,48]. Other studies have shown that metabolic and genotoxic stress-triggered senescence induced by oncogenes is an inherent obstacle to EBV-mediated transformation. As the tumor suppressor p53 is activated under the adverse environment, it leads to the limitation of cell immortalization during early infection and the transformation by EBV [49]. Other studies have shown that oxidative stress is one of the promoters of the EBVs’ induction of B cell immortalization through the regulation of post-transcriptional viruses and cell growth promotion factors [50]. McFadden [49] and Hafez [51] et al. found that EBV-infected cells showed more significant replication stress and DNA damage in early, rapidly proliferating cells than late proliferating cells. Hyper-proliferating B cells showed limited deoxyribonucleic acid triphosphate (dNTP) pools, while late-proliferating and EBV-immortalized lymphoblastoid cell lines showed loss of purine dNTPs. Notably, the supplementation of exogenous nucleosides before the hyper-proliferation stage can significantly enhance B cell immortalization by EBV and alleviate the replication stress. This suggests that the synthesis of purine dNTP is one of the crucial factors in the early stages of EBV-transformed B cells [49,51]. Recent data suggest that EBV controls lymphocyte growth by globally reorganizing host 3D genome architecture to facilitate the expression of key oncogenes [52]. Hence, the concerted activity of EBV-latent proteins in the nucleus contributes to the efficient *in vitro* transformation of primary resting B lymphocytes into immortalized LCLs [53].

So far, LCLs transformed by EBV have been developed as a new method to efficiently immortalize human memory B cells. This has produced a series of important antibodies that have a broad spectrum and neutralizing antibody activity in biology and medicine research and have played an essential role in the prevention and treatment of viral diseases and the design of new vaccines [54]. A fully H1N1 neutralizing antibody 32D6 has been identified by this technique [25]. However, EBV usually infects B cells only in humans and some primates and is highly selective for host cells, limiting its use in other animals.

## 3. Immortalize Human Peripheral B Lymphocytes by Simian Virus 40

Simian virus 40 (SV40) was first identified in rhesus monkey kidney cell cultures used to produce the polio vaccine in 1960 [55]. It is named for the effect on infected cells that produce a number of abnormal vacuoles, as SV40 may promote tumor growth in animal models and induce the transformation of primary cultured human cells [56,57]. SV40 infects rodent or mammalian cells to improve cell immortalization rate and is widely used in experimental models of mammalian cell replication and gene expression [58,59]. SV40 is a double-stranded DNA virus whose genome encodes seven proteins, three structural proteins, and four functional proteins in an overlapping reading frame [58,60]. Structural proteins are VP1, VP2, and VP3, and functional proteins include large T (LT) antigens and small T (ST) antigens essential to the virus life cycle and two small proteins whose functions are unknown [59]. SV40 LT antigen, a multifunctional regulatory protein [61], is classified as a member of the helicase superfamily and has the property of releasing double-stranded DNA and RNA, which play a decisive role in the initiation of virus-transformed cells [62].

There are two main mechanisms by which the SV40 LT antigen immortalizes cells (Figure 2) [63]. One is the activation of E2F-mediated transcription by binding to the Rb-E2F complex. The other is to inhibit p53 activity by blocking p53-dependent transcriptional activation and p53-independent growth arrest. LT antigen binds to the Rb protein to release the inhibition of cell cycle regulation by the Rb-E2F complex, leading to the entry of the S phase with E2F-dependent gene transcription from the growth arrest pathway and resulting in continuous cell proliferation. During the cell transformation by SV40, the LT antigen interacts with the tumor suppressor p53, a transcriptional activator that mediates apoptosis under adverse conditions such as DNA damage, nucleotide deficiency, and abnormal inhibition of Rb protein, thereby transforming cells and extending their lifespan [64,65]. Moreover, the binding of LT to cellular factors, p300/CBP, also prevents apoptosis and leads to the survival of cells independent of direct interaction between LT and p53 [66]. However, ST antigens can enhance cell transformation but are not necessary. In brief, LT antigens and ST antigens maintain cell transformation phenotypes together [67].

Many studies on SV40 immortalized cells, such as cardiomyocytes, nerve cells, macrophages, and venous endothelial cells, have been reported. Theoretically, SV40 can immortalize B cells, but only a few studies on B cell proliferation and immortalization *in vitro* have been reported. Franca Nneka Alarib et al. have shown that the SV40 virus can effectively infect isolated normal human B lymphocytes from PBMCs [65], prolong the lifespan of B lymphocytes, and promote cell proliferation in the transformation process. SV40 LT antigen, which mediates cell immortalization by Rb-E2F and P53, is the primary mechanism for prolonging B cell survival. In Kanki’s study, the SV40 gene was packaged into plasmids, immortalizing isolated human peripheral blood B lymphocytes into a permanently proliferative cell line. The cell line produces and secretes IgG antibodies [68]. At present, there are few studies on the immortalization of B cells in other species, but the transformation of human peripheral blood B cells by SV40 is a valuable case study.

## 4. Immortalize B Lymphocytes by *In Vitro* Gene Transduction

Lentiviral vectors have been designated as a tool for gene therapy because of their characteristic to induce all types of non-diving or slowly proliferating cells, which makes them very significant for clinical applications [69]. Proto-oncogene Bcl-6 belongs to the B-cell lymphoma family of genes as a factor suppressing apoptosis, which is widely involved in the processes of cell differentiation, activation, cell cycle regulation, etc. Differentially expressed in cell types, Bcl-6 is expressed at high levels only in germinal center (GC) B cells and lymphomas with a germinal center B (GC-B) cell phenotype. Bcl-6 works by inhibiting transcriptional activity through the interaction with transcription factors; even the binding site of Bcl-6 is far from the initiation site [70]. It inhibits apoptosis and participates in the cell cycle arrest response by directly inhibiting the transcription of the p53 tumor suppressor gene or by binding to the transcriptional activator protein inhibitor of activated STAT2 (PIAS2), thereby inhibiting the activation of the cell-cycle arrest gene p21. Therefore, Bcl-6 may enable GC-B cells to maintain physiological genotoxic stress associated with high proliferation but does not cause p53-dependent or p53-independent growth arrest and apoptotic responses [71]. Studies on mice have shown that Bcl-6 is an essential factor for GC formation and is beneficial to the proliferation of isolated human B cells by inhibiting the differentiation of B cells into plasma cells [72]. Plasma cells express the transcription factor B lymphocytes-induced mature protein-1 (*Blimp-1*), which is necessary for plasma cell differentiation [73], while Bcl-6 ectopic expression can inhibit the expression of *Blimp-1* by binding to Bcl-6 response elements in the *prdm1* gene, thus inhibiting plasma cell differentiation [74]. Other studies have shown that the co-expression of Bcl-xl, Bcl-2, and Mcl1, a variety of Bcl-2 family genes, results in a strong inhibition of apoptosis in Bcl-6 transduced cells. Therein, Bcl-xl, an essential member of the Bcl-2 family that is mainly investigated in tumorigenesis and drug resistance, has the best effect on cell immortalization by inhibiting apoptosis through the interaction with various proteins [75].

Co-cultured with CD40 ligand (CD40L) and interleukin-21 (IL-21), which are produced by follicular helper T cells, the isolated peripheral blood memory B cells transduced by Bcl-6 and Bcl-xl transgenes could be transformed into cell surface B cell receptor (BCR)-positive and Ig-secreting immortalized B cells. Kwakkenbos et al. set up the immortalized B cell culture system with mouse fibroblasts expressing cytokines and CD40L as the feeder layer and forced the expression of Bcl-6 by a retrovirus-mediated method that can prevent B cells from differentiating into plasma cells [76]. When cell death is inhibited, Bcl-6/Bcl-xL-mediated B cells proliferate rapidly, stimulated by a variety of cytokines (including IL-4, IL-10, and IL-21) to immortalize B cells by *in vitro* gene modification. The overexpression of Bcl-6 and Bcl-xl in combination with the CD40L/IL-21 culture system provided immortalized B cells with GC-B cell-like characteristics; thus, a variety of antiviral-specific antibodies were successfully screened, isolated, and generated [77]. Studies have cloned respiratory syncytial virus-specific and influenza-specific B cell lines and used these cells as antibody sources to effectively neutralize the virus *in vivo*. This method provides a new tool not only for studying B-cell biology and signal transduction by antigen-specific B-cell receptors, but also for the rapid preparation of high-affinity human monoclonal antibodies.

In Westerhuis et al.’s study, CD27+ IgG+ memory B cells isolated from peripheral blood by fluorescence-activated cell sorter (FACS) were stimulated with CD40L and interleukin-21 (IL-21) for 36 h and then transfected with retroviruses containing Bcl-6, Bcl-xl, and green fluorescent protein (GFP)-labeled genes. Immortalized B cell lines by transduction could be preserved for a long time, and they expressed cell surface immunoglobulin and cross-neutralizing antibodies of the virus isolated for later use in the culture supernatant [78]. In addition, Kwakkenbos et al. also explored the application of this method in other species and proved that immortalizing B cells by *in vivo* gene modification was applicable to non-human species such as rabbits, mice, and camels [76]. Briefly, Bcl-6 and Bcl-xl were introduced into mouse B cells through retrovirus-mediated gene modification, and rapid cell expansion was observed for one month in the presence of IL-21 and CD40L, which was similar to that of human B cells, indicating that Bcl-6 and Bcl-xl allowed the expansion of B cells in multiple species [72]. In particular, the B-cell transduction rate of rabbits reached 90%, and its growth was very rapid, with an average doubling time of only 18 h, while the doubling time of human cells was 25–29 h. Similar to human B cells, rabbit B cells also secrete immunoglobulin and express BCR on their cell surface; thus, antigen-specific classification and the functional screening of antibodies secreted by immortalized rabbit B cells can be performed (Figure 3).

The gene modification method that forces the expression of Bcl-xl and Bcl-6 could effectively transfect variable cell types of the B-cell lineage, providing a new approach for immortalizing different types of B-cells. The immortalization of B cells is no longer limited to humans and mice, and functional antibodies have been successfully obtained in other animals, such as monoclonal antibodies in non-human primates and monoclonal antibodies in rabbits, which lays the foundation for the development of antibodies with specific functions. On the other hand, immortalized B cells produced by genetic modification are extremely stable. Their ability to proliferate and produce antibodies is unaffected after repeated freezing and thawing. More importantly, immortalized B cells transfected by this method are consistent with the phenotype of plasma cells producing antibodies, expressing the high-affinity surface marker BCR, which provides an effective way to prepare monoclonal antibodies.

## 5. Immortalize B Lymphocytes by Activation of CD40 Signal

As telomere gradually shortens with the progress of cell division and eventually leads to replicative aging characterized by permanent growth cessation [79], differentiated human cells usually have a limited ability to proliferate *in vivo* and *in vitro*. Meanwhile, telomerase can increase the length of telomeres, which increases telomere duplication at the ends of chromosomes. Many studies find that telomerase is only active in germ cells, which can also be induced in some somatic cells, such as CD40 signal-activated B lymphocytes [80]. Moreover, CD40-activated B cells are potent antigen presenting cells, inducing detailed T-cell responses *in vivo* and *in vitro* [81]. Human B lymphocytes activated by CD40 showed vigorous telomerase activity, which was related to the maintenance and extension of telomeres after the activation of GC-B cells [82]. Thus, the longevity of B memory cells is maintained with a stable or increased telomere length, despite multiple cell divisions in the generation of a memory B cell. In the presence of CD40L and cytokines like interleukin (IL)-2, IL-4, IL-10, and IL-21, the human B lymphocyte surface receptor CD40 can be activated and induce the proliferation of B cells for up to ten weeks *in vitro* [82,83,84]. In summary, telomere maintenance by the up-regulation of telomerase activity is a possible mechanism of limited B cell expansion [76].

Wiesner M et al. introduced a method for the conditional immortalization of human B cells by CD40 ligation [83]. This signal combination simulates T-helper cell activation of B cells, and CD40-stimulated B cell culture has been used *in vitro* to stimulate the differentiation of B cells into memory B cells or plasma cells [84]. They modified Banchereau J et al.’s protocol for a long-term culture of human B cell lines dependent on IL-4 and antibodies to CD40 and re-evaluated the conditions for CD40 stimulation of primary human B cells. It was found that cyclosporin A was necessary for immortalization in addition to CD40L and IL-4 stimulation. When re-stimulated every 5–7 days with fresh stimulator cells, B cells multiplied by 370 populations over more than 1650 days, indicating that these B cells from unseparated peripheral blood mononuclear cells (PBMC) were conditioned to immortalize successfully *in vitro* [84]. CD40-stimulated B cell cultures proliferated for a long time with a constant phenotype of activated B cells when modified *in vitro* stimulation conditions were applied. This suggests that a differentiated human B cell immortalization program could be obtained only by exogenous stimulation with regular and repeated CD40 ligand/IL-4 stimulation plus cyclosporin A.

In a long-term study of the phenotypes of immortalized B cells stimulated by CD40, high expression of antigen-presenting molecules, co-stimulating molecules, and adhesion molecules reflected the role of activated helper T cells in B cell activation and antibody production. During long-term culture, the expression level of those molecules remained unchanged, similar to those detected on virus-transformed B lymphocyte cell lines [1]. Furthermore, CD40-stimulated B cells strongly expressed the B cell surface markers CD19, CD20, and CD21, the activated receptor CD40, and the apoptotic signal receptor CD95. In addition, most CD40-B cell cultures also expressed the memory B cell marker CD27 [85]. The expression level of the differentiated marker CD38 was similar to that of peripheral blood B cells *in vitro*, indicating that immortalized B cells generated by this method were consistent with the phenotype of plasma cells and had the potential function of secreting specific antibodies, which provided an effective way for the preparation of monoclonal antibodies. In general, long-term CD40-stimulated B cells exhibited the phenotypic characteristics of activated B cells, and no other cell types were found in their cultures. Conveniently, CD40-stimulated B cell cultures can be established from most healthy adult donors. These B cells have a constant phenotype, do not contain viruses, constantly rely on the CD40 signal, and have constitutive telomerase activity and stable telomere length. In addition, they can be used to amplify antigen-specific cytotoxic T cells and screen-specific antibodies *in vitro*. These studies show that B cells can be immortalized by CD40 stimulation *in vitro* without gene modification. However, relevant applications of this method are not widely used in other species, and further exploration of its morphic mechanism is needed.

The starting point, species, approaches and final products of the human B cell immortalization methods are compiled in Table 1.

## 6. Conclusions

As one of the most important biotechnology products, mAb is widely employed in the diagnosis and treatment of diseases, health diagnosis, and clinical diagnosis of histocompatibility antigen typing for stem cell transplantation, etc. Currently, the US FDA has approved more than 100 mAbs against a wide range of illnesses such as cancer, autoimmune diseases, infectious diseases, and neurological disorders. Human monoclonal antibody development is the future of mAb research, the leading strategy of biotechnology and biomedical industries.

It is theoretically simple and effective to obtain human monoclonal antibodies directly from human B cells that secrete natural antibodies, but technically difficult. The immortalization of human B cells is a meaningful way to produce human monoclonal antibodies efficiently. With the development of biotechnology, different methods of human B cell immortalization are gradually updated, although every method has its own limitations. Further exploration of immortal molecular mechanisms, such as the control of apoptosis and the maintenance of telomere, is an essential prerequisite for selecting immortal methods. From the perspective of methodology, in-depth research on gene-modified vectors and specific transformed virus vectors is the precondition for the immortalization of human B cells. Excitingly, antibodies produced by spontaneous immortalized human B cells are simple and safe in applications. These foregoing studies have explored the preparation of human antibodies with various specific functions by human B cell immortalization, which are valuable tools for the development of monoclonal antibodies. In the future, this technology will be used as a significant model system providing the basis of personalized medicine.

## Figures and Tables

**Figure 1 microorganisms-11-02936-f001:**
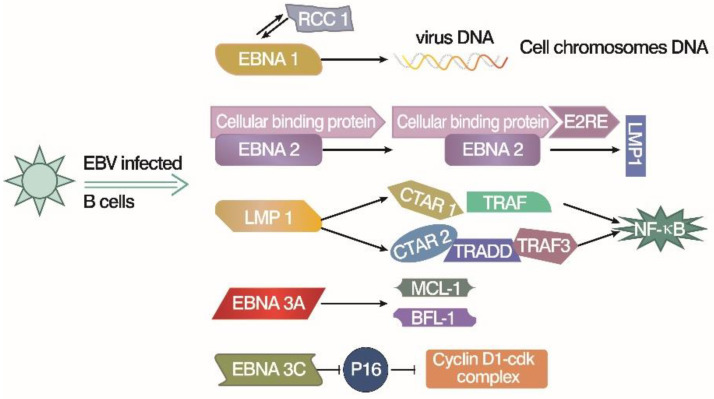
EBV latent gene products work on B cell immortalization [26,27,28,29,30,31,32,33,34,35,36,37]. *EBNA1* may interact with RCC1, which leads the viral DNA onto the chromosomes and promotes virus replication. *EBNA2* upregulates the other viral EBNA genes, like *LMP1*, by forming a complex with E2RE and cellular binding protein. LMP-1 activates NF-KB directly or indirectly through interaction with key intermediate proteins (TRAF and TRADD, etc.). *EBNA3A* upregulates the *MCL-1* and *BFL-1* proteins, which are related to cell death, or apoptosis. *EBNA3C* inhibits p16 to upregulate the Cyclins and B cell proliferation.

**Figure 2 microorganisms-11-02936-f002:**
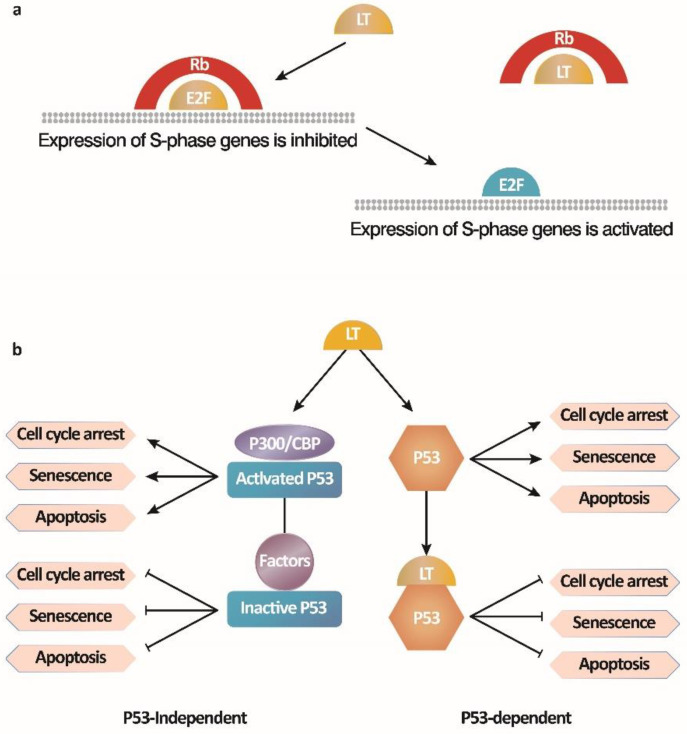
Two mechanisms by which the SV40 LT antigen immortalizes B cells [63]. (**a**) SV40 LT antigen activates E2F-mediated transcription by binding to the Rb-E2F complex; (**b**) SV40 LT antigen inhibits p53 activity by blocking p53-dependent transcriptional activation and p53-independent growth arrest, preventing B cell apoptosis.

**Figure 3 microorganisms-11-02936-f003:**
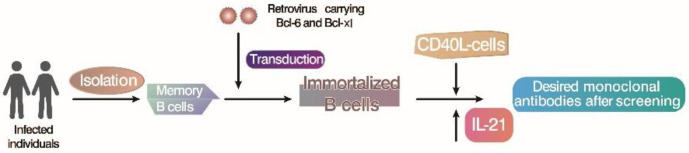
The establishment and maintenance of immortalized B cells through *in vitro* genetic modification [76]. Isolate B cells from selected individuals, stimulate and transduce with retroviruses containing Bcl-6 and Bcl-xl, and then, in the presence of interleukin IL-21, successfully transduced immortalized B cells expressing CD40L continue to expand on fibroblasts; screen B cell clones to produce the desired antibody.

**Table 1 microorganisms-11-02936-t001:** Summary for human B cell immortalization methods.

Methods	Starting Point	Species	Approaches	Way to Maintain Cell Cultures	Final Products	References
EBV	Isolated mononuclear leukocytes	Human, some primates	Transformation	Latent genes, p53 tumor suppressor, telomerase, etc	LCLs	[23,24,25,26,27,28,29,30,31,32,33,34,35,36,37,38,39,40,41,42,43,44,45,46,47,48,49,50,51,52,53]
SV40	Isolated human B lymphocytes	Mammalian	Infection	LT antigen regulated E2F and p53 down-stream gene, ST antigens	Permanently proliferative cell line	[65,68]
Gene transduction	Isolated human B cells	Multiple species	*In vitro*	Ectopic expression of Bcl-6 and Bcl-xl with proper feeder cells	Variable immortalized B-cell lineage	[72,75]
Activation of CD40 signal	PBMC	Human	Exogenous stimulation	Regular and repeated CD40 ligand/IL-4 stimulation plus cyclosporin A	Immortalized B cells with phenotypic characteristics of activated B cells	[82,83,84]

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
