# Peer review of "The Research Progress on Immortalization of Human B Cells"

_microorganisms, 2023, doi:10.3390/microorganisms11122936_

Round 1
Reviewer 1 Report
Comments and Suggestions for Authors
They are included in the attached PDF file

No further comments
Author Response
Many thanks for your comments.

Reviewer 2 Report
Comments and Suggestions for Authors
This is a very nice and well-written review about how to immortalize B lymphocytes. After introducing several ways to immortalize cells in general and B cells in particular, the manuscript focuses on several specific ways to do so. One can feel that the manuscript focuses on antibody-producing B cells as they have a commercial and clinical interest. There are several minor points to consider regarding this manuscript.
1. Abelson leukemia virus-transformed progenitor or pre-B cells (vAbl cells) could be introduced and mentioned, either in the core part of the manuscript or in the introduction. These cells were used a lot in research and could be listed at minimum as a general way to make the cell line, especially with Bcl2+ transgene, which induces cell survival.
2. Line 146. "2. Immortalize human peripheral blood B lymphocytes by Epstein-Barr virus" This section is about the SV40 virus, however, the title is probably copied from the previous section. Should it be section 3? And another title?
3. Line 193. Is it section 4?
4. Line 273. Section 5?
5. While mentioning different approaches to immortalizing B cells, and different protocols, when available, it would be beneficial to mention the starting point of immortalization. Was it always PBMC? Was it an isolated B-cell population? For some methods it is done, is it possible to do it for all?
6. Line 327. Conclusion - section 6?
7. It looks useful to have a table summarizing B cell immortalization methods, described in the manuscript, including various parameters, e.g., starting point,(PBMC, isolated fractions), species (human, mice, etc), way to immortalize, ways to maintain cell cultures, final products, references, etc. This is just an example, however, the table would improve the presentation of the results, especially the summary part.
Author Response
Many thanks for your comments.

Reviewer 3 Report
Comments and Suggestions for Authors
Reviewer comments
Manuscript: microorganisms-2686320 - The Research Progress in Immortalization of B Cells
The authors reviewed the progress of research in the field of B-cell immortalization. Immortalization of B cells, maintaining constant growth characteristics and antibody expression of B cells in vitro, is very important for the development of antibody drugs and products for diagnostic and therapeutic purposes. Methods for immortalizing B cells include Epstein-Barr virus (EBV) transformation, Simian virus 40 (SV40) infection, in vitro genetic modification, CD40 activation, etc. Immortalized B cells are very efficient in producing monoclonal antibodies (mAbs), and Antibodies produced in this way can overcome immune rejection caused by heterologous antibodies. It is an effective method for producing monoclonal antibodies and an important method for developing therapeutic monoclonal antibodies. This article reviews the progress of research in the field of B cell immortalization, its methods and future direction, as it is a powerful tool for the development of monoclonal antibody technology.
The data analysis methods are correct.
The English of the text is well written and well readable but needs additional checking with a professional translator.
The uniqueness of the text is more than 90% by AntiPlagiarism.NET.
The text contains some misspellings and typos. Also need to expand the part of the discussion.
There are some comments and questions:
1) What is novelty of this manuscript? Everything about immortalization of B Cells is known now.
2) The authors have not described an approach to immortalize cells through expression of the telomerase reverse transcriptase (TERT) protein, especially for cells that are most affected by telomere length, such as human cells (Lundberg et al., 2000; Fridman and Tainsky, 2008). Please add this method if possible.
3) Add to the References:
Lundberg AS, Hahn WC, Gupta P, Weinberg RA. Genes involved in senescence and immortalization. Curr Opin Cell Biol. 2000 Dec;12(6):705-9. doi: 10.1016/s0955-0674(00)00155-1. PMID: 11063935.
and
Fridman AL, Tainsky MA. Critical pathways in cellular senescence and immortalization revealed by gene expression profiling. Oncogene. 2008 Oct 9;27(46):5975-87. doi: 10.1038/onc.2008.213. Epub 2008 Aug 18. PMID: 18711403; PMCID: PMC3843241.
4) The manuscript reviewed recent publications but there is not recent papers of 2023.
For example: Zhao B. Epstein-Barr Virus B Cell Growth Transformation: The Nuclear Events. Viruses. 2023 Mar 24;15(4):832. doi: 10.3390/v15040832. PMID: 37112815; PMCID: PMC10146190.
and
Wang C, Liu X, Liang J, Narita Y, Ding W, Li D, Zhang L, Wang H, Leong MML, Hou I, Gerdt C, Jiang C, Zhong Q, Tang Z, Forney C, Kottyan L, Weirauch MT, Gewurz BE, Zeng MS, Jiang S, Teng M, Zhao B. A DNA tumor virus globally reprograms host 3D genome architecture to achieve immortal growth. Nat Commun. 2023 Mar 22;14(1):1598. doi: 10.1038/s41467-023-37347-6. PMID: 36949074; PMCID: PMC10033825.
5) The Conclusions is weak, please improve it and add importance of your manuscript for development of science and medicine.
6) The manuscript is important and interesting, well english written, no typos and mistakes. The Introduction need in additional recent information, should be little extended.
Please improve the manuscript according to the above comments.
Comments on the Quality of English LanguageThe quality of English is well. Minor editing of English language required
Author Response
Many thanks for your comments.

Round 2
Reviewer 1 Report
Comments and Suggestions for Authors
No further comments